# *Lacticaseibacillus paracasei* GM-080 Ameliorates Allergic Airway Inflammation in Children with Allergic Rhinitis: From an Animal Model to a Double-Blind, Randomized, Placebo-Controlled Trial

**DOI:** 10.3390/cells12050768

**Published:** 2023-02-28

**Authors:** En-Kwang Lin, Wen-Wei Chang, Jhih-Hua Jhong, Wan-Hua Tsai, Chia-Hsuan Chou, I-Jen Wang

**Affiliations:** 1Institute of Clinical Medicine, National Yang Ming Chiao Tung University, Taipei 112304, Taiwan; 2Division of Colorectal Surgery, Department of Surgery, Wanfang Hospital, Taipei Medical University, Taipei 110301, Taiwan; 3School of Biomedical Sciences, Chung Shan Medical University, Taichung 402306, Taiwan; 4Department of Medical Research, Chung Shan Medical University Hospital, Taichung 402306, Taiwan; 5Department of Medical Research, Hsinchu MacKay Memorial Hospital, Hsinchu 300044, Taiwan; 6Research and Development Department, GenMont Biotech Incorporation, Tainan 741014, Taiwan; 7Department of Pediatrics, Taipei Hospital, Ministry of Health and Welfare, New Taipei 242033, Taiwan; 8School of Medicine, National Yang Ming Chiao Tung University, Taipei 112304, Taiwan; 9College of Public Health, China Medical University, Taichung 406040, Taiwan; 10National Institute of Environmental Health Sciences, National Health Research Institutes, Miaoli 350401, Taiwan

**Keywords:** allergic airway inflammation, animal model, allergic rhinitis, clinical trial, probiotics

## Abstract

*Background:* Probiotics may facilitate the clinical management of allergic diseases. However, their effects on allergic rhinitis (AR) remain unclear. We examined the efficacy and safety of *Lacticaseibacillus paracasei* GM-080 in a mouse model of airway hyper-responsiveness (AHR) and in children with perennial AR (PAR) by using a double-blind, prospective, randomized, placebo-controlled design. *Methods:* The production of interferon (IFN)-γ and interleukin (IL)-12 was measured by using an enzyme-linked immunosorbent assay. GM-080 safety was evaluated via the whole-genome sequencing (WGS) of virulence genes. An ovalbumin (OVA)-induced AHR mouse model was constructed, and lung inflammation was evaluated by measuring the infiltrating leukocyte content of bronchoalveolar lavage fluid. A clinical trial was conducted with 122 children with PAR who were randomized to receive different doses of GM-080 or the placebo for 3 months, and their AHR symptom severity scores, total nasal symptom scores (TNSSs), and Investigator Global Assessment Scale scores were examined. *Results:* Among the tested *L. paracasei* strains, GM-080 induced the highest IFN-γ and IL-12 levels in mouse splenocytes. WGS analysis revealed the absence of virulence factors or antibiotic-resistance genes in GM-080. The oral administration of GM-080 at 1 × 10^7^ colony forming units (CFU)/mouse/day for 8 weeks alleviated OVA-induced AHR and reduced airway inflammation in mice. In children with PAR, the oral consumption of GM-080 at 2 × 10^9^ CFU/day for 3 months ameliorated sneezing and improved Investigator Global Assessment Scale scores significantly. GM-080 consumption led to a nonsignificant decrease in TNSS and also nonsignificantly reduced IgE but increased INF-γ levels. *Conclusion:* GM-080 may be used as a nutrient supplement to alleviate airway allergic inflammation.

## 1. Introduction

Allergic rhinitis (AR) and asthma, which occur in the upper and lower airways, respectively, and their co-occurrence are noted in >40% of patients with allergic airway inflammation [1]. T helper 2 (Th2) cells are considered a critical factor in the development of allergic airway inflammation. Animal models have revealed that the transfer of antigen-specific Th2 cells into naïve mice followed by challenge with the specific antigen through inhalation induces asthmatic responses, such as airway hyper-responsiveness (AHR), eosinophilic inflammation, and mucus hyperproduction [2,3]. The airway expression of interleukin (IL)-4, IL-5, and IL-13 released by Th2 cells is a crucial mediator [4,5,6]. However, the major limitations related to biological drugs for allergic airway inflammation, such as dupilumab, include high costs and the loss of responsiveness over time [7]. Thus, low-cost alternative agents for allergic airway inflammation should be developed.

Although animal and human studies [8] have indicated the benefits of probiotics in allergic disease treatment, medical societies hold a conservative attitude toward probiotic use. The World Allergy Organization [9] states that despite the insufficient volume of data on the relevant probiotic strains and dosages, the application of probiotics in pregnant and breastfeeding women, as well as in infants with a family history of allergic disease, particularly eczema, may have considerable effects [10]. The term “pharmabiotics” has been proposed, and the underlying immunomodulatory pathway is of interest to several researchers and clinicians [11]. In a mouse model of house dust mite-induced asthma, Voo et al. found that combination treatment with *Lacticaseibacillus rhamnosus* and corticosteroid reduced AHR, serum IgE levels, and Th2 cytokines [12]. Lan et al. reported that the oral administration of *Lactiplantibacillus plantarum* CQPC11 strain to an ovalbumin (OVA)-induced asthmatic mouse model decreased OVA-specific IgE or IgG1 in sera and proinflammatory cytokines in bronchoalveolar lavage fluid (BALF) [13]. Yan et al. reported a meta-analysis of the beneficial effects of probiotics on AR by collecting 30 randomized controlled trials. Their results revealed that the consumption of probiotics improved the scores of Rhinitis Quality of Life and Rhinitis Total Symptom but not immunological parameters, including blood eosinophil count or total and antigen-specific serum IgE levels [14]. Given the unequal effects of different probiotic species or strains in treating allergic AHR [15,16], investigating the efficacy of different probiotic strains in cell models, animal models, and clinical trials before their use as food supplements is essential.

Here, we selected *Lacticaseibacillus paracasei* strain GM-080 (LP-33, BCRC 910220, CCTCC M 204012), which has been reported to alleviate allergic dermatitis in infants [17], grass pollen-induced persistent AR in adults [18], and perennial AR (PAR) in infants [19], as the main target probiotic. We examined its therapeutic effects in an asthma mouse model and a double-blind, prospective, randomized, placebo-controlled study on children with PAR.

## 2. Materials and Methods

### 2.1. L. paracasei Strain and Cell-Wall Component Preparation

*L. paracasei* strains (GM-080 and GM-2–GM-23; Figure 1A) were obtained from GenMont Biotech (Tainan, Taiwan). Bacterial cells were cultured in MRS broth overnight, then resuspended in sterile phosphate-buffered saline (PBS), and subsequently diluted to 4 × 10^7^ cells/mL. Lipoteichoic acid (LTA) and peptidoglycan (PGN) were purified from live concentrated GM-080 (1 × 10^11^ colony forming units [CFU]/mL) as previously reported [20].

### 2.2. Mouse Splenocyte Stimulation and Enzyme-Linked Immunosorbent Assay–Based Detection of Cytokines

The mouse spleen was excised, passed through a 45 µm cell strainer (BD Biosciences, Franklin Lakes, NJ, USA) for conversion into a single-cell suspension, then subjected to red blood cell lysis by using RBC lysis buffer (eBiosciences Inc., San Diego, CA, USA). Isolated splenocytes were seeded into a 96-well-plate at a cell density of 4 × 10^5^ cells/well and incubated with 2 µg/mL ConA, 1 µg/mL lipopolysaccharide (LPS), or GM-080 with different multiplicities of infection (MOIs) separately for 48 h. Cell culture supernatants were then harvested, and the presence of cytokines (IL-5, IL-12, and IFN-γ) was detected by utilizing commercial enzyme-linked immunosorbent assay (ELISA) kits (Cat. No. 555256 for mouse IL-12[P70] and Cat. No. 555138 for mouse IFN-γ; purchased from BD Biosciences, Franklin Lakes, NJ, USA).

### 2.3. Antimicrobial Susceptibility Profiling

Antimicrobial susceptibility was determined by using the broth microdilution method and lymphocyte separation medium (LSM, including 90% IST medium [Cat. No. CM0473; Oxoid, Basingstoke, Hampshire, UK] and 10% MRS medium (Cat. No. 288130; Difco Laboratories Inc., Franklin Lakes, NJ, USA) in accordance with the guidelines of the Quality and Standards Authority of Ethiopia (ES ISO10932:2012). Twofold dilutions of clinically relevant antibiotics (clindamycin, chloramphenicol, erythromycin, gentamicin, kanamycin, streptomycin, tetracycline, and ampicillin, all from Sigma-Aldrich, Saint Louis, MO, USA) were prepared in LSM. Approximately 50 μL of 6 × 10^5^ CFU/mL *L. paracasei* cells were loaded into a 96-well plate, followed by 50 μL of multiple antibiotics diluted in LSM. The plate was incubated under anaerobic conditions at 37 °C for 16–24 h. Minimum inhibitory concentrations (MICs) were defined as the lowest concentrations of antibiotics at which the growth of *L. paracasei* was completely inhibited. Strains were classified as susceptible or resistant by using the microbiological cutoffs established by the European Food Safety Authority (EFSA) [21].

### 2.4. DNA Extraction, Whole-Genome Sequencing, and Hybrid Genome Assembly

We established a whole-genome sequencing (WGS) assembly pipeline in accordance with a previously reported procedure [22] with minor modifications. In brief, genomic DNA was fragmented through ultrasonication by using Covaris S2 (Covaris, Woburn, MA, USA). Indexed polymerase chain reaction-free library construction was performed by using the multiplexed high-throughput sequencing TruSeq DNA Sample Preparation Kit (Illumina, Inc., San Diego, CA, USA) in accordance with the manufacturer’s protocols with minor modifications. The GM-080 genome was deeply sequenced through Nanopore GridIon long-read sequencing with 175-fold coverage and whole-genome shotgun sequencing by using 2 × 250-bp paired-end sequencing at 125-fold coverage on a MiSeq platform and hybrid-assembled on a MaSuRCA v3.3.1 assembler [23]. Benchmarking Universal Single-Copy Orthologs (version 4.0.0) [24] was applied for genome completeness assessment through comparison with the lactobacillales_odb10 gene database. We deposited all the sequencing data in GenBank under BioProject ID no. PRJNA824946.

### 2.5. Annotation of Protein-Coding Genes, Virulence Factors, and Antibiotic Resistance

The protein-coding genes in the GM-080 genome were annotated by using Prokka [25]. On the basis of the risk assessment of potential genes of concern for microorganisms to be used in the food chain established by EFSA [26], virulence factors in the genome were separately identified by running BLAST against the virulence factor database (VFDB) [27] by using the following criteria: query sequence hits with identity ≥80%, alignment coverage > 70%, and E-value < 10^−30^. Antibiotic-resistance genes were predicted through a BLAST search against both the Comprehensive Antibiotic Resistant Database (CARD) [28] and ResFinder (version 4.1) database [29] by using the same criteria. Whole-genome average nucleotide identity was computed by OrthoANI [30]. GM-080 phylogeny and other related genomes were reconstructed using MEGA X [31]. After annotation, the circular genome atlas was generated using the Circos visualization tool [32].

### 2.6. OVA-Induced AHR Mouse Model

Seven-week-old female BALB/c mice were purchased from The Experimental Animal Facility of the College of Medicine, National Taiwan University (Taipei, Taiwan). Allergic airway inflammation was induced with an intraperitoneal injection of 50 µg of OVA (purchased from Sigma-Aldrich) mixed with the Th2-adjuvant aluminum hydroxide on day 0 and was followed by the administration of 25 µg of OVA on days 14, 28, 42, and 56 to sensitize the mice and then by intranasal challenge with OVA (100 µg) on days 67 and 68. Sera were collected from the retro-orbital sinus on the day before the first OVA sensitization; on post sensitization days 35, 49, and 63; and on the day of sacrifice. 

### 2.7. ELISA Determination of OVA-Specific Immunoglobulins

A 96-well plate was coated with OVA at 1 µg/well and then incubated with blocking solution (1% bovine serum albumin in PBS buffer) at room temperature. The wells were added with mouse serum followed by antibodies diluted with the blocking solution (1:50, 1:1000, and 1:10,000 dilutions for IgE, IgG2a, and IgG1, respectively) and incubated at 4 °C overnight. Biotin-conjugated antibodies against mouse IgE, IgG1, or IgG2a (BD Biosciences) were added after the wells were washed with 0.05% Tween-20/PBS buffer. Then, the plate was incubated at room temperature for 2 h. Next, streptavidin-conjugated horseradish peroxidase was added to the wells. 3,3′,5,5′-Tetramethylbenzidine substrate was used for color development. After the termination of the reaction by using 2 N H_2_SO_4_, absorbance was measured at 450 nm on a VERSAmax microplate reader (Molecular Devices, San Jose, CA, USA).

### 2.8. AHR Determination

In mice, the development of AHR was determined on a Buxco system (Biosystem XA; Buxco Electronics, Sharon, CT, USA). The enhanced pause (Penh) values were calculated with the following formula: (*pause* × *PIF*)/*PEF*, with *pause* being (*T*e − *T*r)/*T*r. Here, *PIF* is the peak inspiratory flow, *PEF* is the peak expiratory flow, *T*e is the expiratory time, and *T*r is the relaxation time. First, we recorded the Penh of the mice under awake conditions by using a whole-body plethysmograph chamber for 3 min with normal saline vaporization. Next, the mice received methacholine-containing aerosols at increasing concentrations of 6.25, 12.5, 25, and 50 mg/mL for 3 min. The Penh data are presented as the relative increase in Penh after normalization with Penh after PBS inhalation.

### 2.9. BALF Collection and White Blood Cell Count

BALF was collected through lung flushing with HBSS buffer containing 2% bovine serum albumin three times by using a trachea cannula (Angiocath, BD Biosciences). BALF cells were then collected through centrifugation at 300× *g* for 5 min and spanned on slides by using a cytocentrifuge (Thermo Fisher Scientific, Waltham, MA, USA) and then subjected to Liu’s staining. Monocyte, lymphocyte, neutrophil, and eosinophil numbers within the cell pellet were then counted with a 100× objective lens. The data are presented as the average cell count in at least five fields for each sample. BALF eotaxin and IL-5 levels were determined by using commercial ELISA kits (BD Biosciences).

### 2.10. Patient Recruitment

PAR was diagnosed in accordance with the definition of Saleh and Durham, i.e., symptoms lasting more than 4 days per week and illness duration lasting more than 4 weeks [33]. In total, 156 patients aged 5–16 years old were recruited from the pediatric outpatient clinics of two hospitals (MacKay Memorial Hospital, Taipei, Taiwan and Chang Gung Children’s Hospital, Taipei, Taiwan) for eligibility assessment in our clinical trial. However, 34 patients were excluded because they withdrew without signing the written informed consent form or because they fulfilled the exclusion criteria. The inclusion criteria were (i) age = 5–16 years; (ii) PAR for ≥1 year; (iii) positive on any one of the following tests within 12 months: skin-prick test reaction (wheal size at least 3 mm larger than that made in the diluent control group) or allergic reaction examined by the methods of Pharmacia-CAP or multiple allergens simultaneous test; and (iv) mean total nasal symptoms score (TNSS) of no less than 5 throughout the screening period (at least 4 days) and TNSS on the day before day 0 (first dosing day) visit of no less than 5. Subjects were excluded if they (i) had clinically significant abnormalities in laboratory results as determined during 14 days prior to visit 1 or during the baseline period by the investigator; (ii) had acute or significant chronic sinusitis, severe persistent asthma, congenital immunodeficiency, neuropsychiatric disorders, immune-compromised massive wounds in the oral cavity, use of rhinitis medications, and chronic use of tricyclic antidepressants; (iii) need to take prohibited medications during the study or took the medications within 30 days prior to the screening visit, including parenteral or oral corticosteroids, nasal corticosteroids, topical flurandrenolide, topical clobetasol propionate, topical halobetasol propionate, astemizole, ketotifene, nedocromil or sodium cromoglycate, loratadine, cetirizine, antileukotrienes, other H1 antihistamines, nasal decongestant, or any food supplements including *L. paracasei*; (iv) were undergoing desensitization therapy within 3 months prior to the screening visit or with vasomotor rhinitis; (v) participated in an investigational drug trial within 4 weeks before entering this study; (vi) were pregnant, lactating, or planning to become pregnant; and (vii) had any other serious disease considered by the investigator not in the condition to enter the trial. The trial was approved by the Joint Institutional Review Board (c/o Taipei Veterans General Hospital, Taipei City, Taiwan) with the reference number 05-016-A on the date of 13 November 2006 and complied with the principles of the Declaration of Helsinki. The trial was registered on ISRCTN Registry (ISRCTN14829274, https://www.isrctn.com/ISRCTN14829274, accessed on 28 October 2022). All the eligible patients and their parents or guardians were provided verbal and written information regarding the study and provided written informed consent.

### 2.11. Randomized, Double-Blind, Placebo-Controlled Trial Design

The randomized, double-blind, placebo-controlled trial on PAR was designed in accordance with the guidance for developing drug products for AR treatment published by the Division of Pulmonary, Allergy, and Rheumatology Products in the Center for Drug Evaluation and Research at the Food and Drug Administration (April 2000 version). All patients who met the eligibility criteria were randomized to either the probiotic-treated or control group. In total, 137 patients were double-blinded and randomized to four groups, namely, one placebo group and three groups that received live GM-080 at different dosages: group A (2 × 10^8^ colony forming units [CFU]/day); group B (2 × 10^9^ CFU/day), and group C (1 × 10^10^ CFU/day). All treatments were administered for 3 months. Considering the key factors that could potentially influence treatment outcomes, randomization was stratified by age, sex, and AR severity. All probiotic capsules were supplied and stored at less than 4 °C with cGMP facilities. The demographic characteristics of the patients were collected at baseline by using questionnaires. The primary outcome was the change in AR severity after 3 months of intervention compared with the AR severity at baseline. The patients visited the hospital for data collection six times: at screening (V1) and in treatment weeks 0 (V2), 2 (V3), 4 (V4), 8 (V5), and 12 (V6). TNSS was used to evaluate the severity of main AR symptoms; here, the higher TNSS, the more severe the AR symptoms. Generalized estimating equations (GEEs) [34] were used for a within-subject covariance structure evaluation. The Investigator Global Assessment Scale score at V3–V6 was evaluated by the trial investigator to assess the overall improvement of the participants after the treatment, which was divided into four levels: complete relief (4 points), partial relief (3 points), no relief (2 points), or worse (1 point). All parents were contacted 1 month after treatment to determine whether they observed a relapse after treatment interruption. Changes in severity scores in the groups were evaluated at each visit. The secondary outcomes were changes in total serum IgE and INF-γ levels observed in months 0 and 3. Only 122 patients (28, 31, 31, and 32 in group A, group B, group C, and the placebo group, respectively) were included in the final analysis due to loss of follow-up.

### 2.12. Skin Prick Tests and Serum Biomarkers

Skin prick tests were performed with commercial allergen extracts of egg, milk, crab, mite, cockroach, and animal dander (ALK-Abell & Oacute, Round Rock, TX, USA). Skin reactivity to allergen sensitization was classified into four grades as previously described [35]. Serum total IgE levels were measured by the Division of Laboratory Medicine of MacKay Memorial Hospital (Taipei City, Taiwan) and the Division of Laboratory Medicine of Chang Gung Memorial Hospital, Taipei (Taipei City, Taiwan). Serum IFN-γ levels were measured by using ELISA kits (BD Biosciences).

### 2.13. Statistical Analysis

The data from in vitro and OVA-induced airway inflammation mouse model experiments were presented as means ± standard deviations and were analyzed for differences between groups by using one-way analysis of variance (ANOVA), followed by Tukey’s Honestly Significant Difference test. The chi-square test was conducted to compare clinical symptoms. Intragroup comparisons for TNSSs and blood biomarkers at baseline and 3 months after treatment commencement were carried out by using paired *t*-tests. Intergroup comparisons among the four groups were performed by using ANOVA. The differences in TNSSs at the six visits among the four groups were also evaluated by applying a mixed model with adjustment for potential confounders. All children who completed the study were included in an intention-to-treat analysis regardless of their compliance. All tests assumed a two-sided alternative hypothesis with a significance level of 0.05. All analyses were conducted by using SAS (version 9.1; SAS Institute, Cary, NC, USA).

## 3. Results

### 3.1. GM-080 Induces Th1 Cytokine Production in Mouse Splenocytes

The Th1 cytokines of IFN-γ or IL-12 have been demonstrated to suppress AHR in response to allergens [36,37]. We used mouse splenocytes as a cell model and evaluated Th1 cytokine (IFN-γ and IL-12) production in the cell supernatant after *L. paracasei* exposure to identify the potential *L. paracasei* strains with antiallergic effects. By using ConA or LPS as the positive control (Figure 1 and Figure 2A,F), we observed that out of all the 23 *L. paracasei* strains that were screened, GM-080 demonstrated the strongest IFN-γ (Figure 1B) and IL-12 (Figure 1C) induction activity, whereas BCRC 16100 demonstrated the lowest induction activity. We examined the Th1 cytokine induction ability of live GM-080 and the derived LTA or PGN in mouse splenocytes. The best IFN-γ (Figure 2B) and IL-12 (Figure 2G) induction activity was demonstrated by live GM-080 at MOI = 10 and 0.1, respectively; IL-12 induction was greater because of live GM-080 than because of LPS (Figure 2F). LTA and PGN could induce IFN-γ production with the highest extent of induction at a concentration of 1 mg/mL (Figure 2C,D, respectively). These data suggest that GM-080 has the potential to alleviate allergic conditions.

### 3.2. WGS Analysis Revealed That GM-080 Is a Safe L. paracasei Strain 

We performed WGS analysis on GM-080 by using next-generation sequencing. The genetic organization of GM-080 was illustrated in a genome atlas created by using Circos (Figure 3A). One CRISPR locus, one bacteriocin gene, and six prophage-like clusters (Appendix A) were observed in the GM-080 genome, with no plasmid being identified (Figure 3B). We next examined the determinants of putative antibiotic resistance and virulence factors to evaluate the safety of GM-080 on the basis of the WGS analysis guidelines for probiotics safety provided by EFSA [26]. Our in silico analysis demonstrated no putative antibiotic resistance or virulence factor genes in the GM-080 chromosome. We further examined the safety of GM-080 by using antimicrobial susceptibility profiling in accordance with the guidelines established by QSEA (ET ISO 10932). All MICs of the tested antibiotics to GM-080 were below the cutoffs suggested by EFSA (Appendix A). These data suggest that GM-080 is safe for consumption as a probiotic.

### 3.3. GM-080 Genome Contains Immunosuppressive Motifs and CpG-Containing Oligonucleotides That Induce Th1 Cytokines in Mouse Splenocytes

We compared the genomes of GM-080 and BCRC 16100 to identify the potential functional genes in GM-080. As indicated by the phylogenetic tree (Figure 3C), GM-080 was clustered with BCRC 16100 [22]; however, BCRC 16100 displayed a lower immunomodulation capability than GM-080 (Figure 1). Table 1 shows that the functional categories of genes in GM080 and BCRC 16100 were highly similar. The unique genes in GM-080 are summarized in Appendix A. Immunosuppressive motifs (IMs) in *Lactobacillus* spp. may participate in ameliorating allergic conditions through their anti-inflammatory activity [38]. In addition, the activation of toll-like receptor 9 by CpG-containing oligonucleotides (ODNs) can shift the immune dominance from Th2 to Th1, which may also facilitate the alleviation of allergic diseases [39]. We analyzed well-known IM and CpG ODNs (Appendix A) in two strains and next synthesized eight candidate IM sequences in accordance with the genome sequences of GM-080 (IM4–IM7) and BCRC 16100 (IM8–IM11) and used them to stimulate mouse splenocytes. Although none of the IMs induced IFN-γ production, IM5 from the GM-080 genome exhibited capability for IL-12 induction activity (Table 2). We also synthesized six CpG-containing ODNs on the basis of the genome analysis of GM-080 (ODN1 or ODN2) and BCRC 16100 (ODN3–ODN6) and used them to stimulate mouse splenocytes. None of the BCRC 16100 ODNs exhibited IL-12 induction activity, but both ODNs from GM-080 (ODN1 and ODN2) induced IFN-γ and IL-12 secretion (Table 2). Therefore, we hypothesized that among *L. paracasei* strains, GM-080 would be a probiotic with antiallergic activity.

### 3.4. Orally Gavage GM-080 Alleviates OVA-Induced Allergic Airway Inflammation in Mice

We established an OVA-induced AHR mouse model by using the protocol shown in Figure 4A. Oral GM-080 treatment was administered from days 7 to 63 at the dose of either 2 × 10^6^ CFU/mouse/day (low-dose group) or 1 × 10^7^ CFU/mouse/day (high-dose group). Blood samples were collected on days 63 (4 days before intranasal challenge with OVA) and 70 (the day of sacrifice). After the intranasal challenge with OVA, the Penh value elevated as the methacholine dose was increased, indicating the occurrence of AHR. Oral GM-080 alleviated AHR in a dose-dependent manner (Figure 4B). We also observed that the oral administration of high-dose GM-080 significantly reduced the numbers of eosinophils, neutrophils, lymphocytes, and monocyte numbers in BALF, whereas that of low-dose GM-080 significantly reduced the numbers of neutrophils (Figure 5B) and monocytes (Figure 5D). The expression of eotaxin in BALF was reduced by oral GM-080 (Figure 5E). In addition, the IL-5 levels in BALF were reduced by oral GM-080 at low and high dosages (Figure 5F). Next, we found that oral GM-080 at a low dose significantly reduced anti-OVA IgE (Figure 6A) and anti-OVA IgG2a (Figure 6C) levels, whereas that at a high dose significantly reduced only anti-OVA IgG2a levels (Figure 6C) but not anti-OVA IgG1 levels (Figure 6B). Moreover, oral GM-080 reduced ConA- (Figure 6E) and OVA- (Figure 6F) induced IL-5 production in mouse splenocytes. These data demonstrate that the oral administration of GM-080 can ameliorate allergic airway inflammation in OVA-induced AHR mice.

### 3.5. GM-080 Alleviates PAR in Children

Finally, we conducted a double-blind, randomized, placebo-controlled trial to investigate the beneficial effects of GM-080 in children with PAR. The study design was summarized in Figure 7 and their baseline demographic characteristics were presented in Table 3; no significant differences were noted among the patient groups. We first evaluated the effects of probiotics on AR symptom severity scores. Only the sneezing subscale scores significantly decreased, particularly in group B (in which 2 × 10^9^ CFU/day GM-080 was administered; Table 4). Scores for other symptoms such as rhinorrhea, nasal pruritus, and nasal congestion did not improve in groups A and C compared with the placebo group. By contrast, these scores significantly improved in group B over time (Table 4).

We next examined the effects of GM-080 on patients’ quality of life by using TNSSs. TNSSs are a convenient tool for symptom description and the assessment of functional problems (physical, emotional, social, and occupational) associated with AR. We noted a nonsignificant decrease in TNSSs over time in groups A, B, and C (Table 4). In our GEE model, TNSSs after the five visits among our four groups exhibited no significant differences (Table 5). Investigator Global Assessment Scale scores were higher in the groups that were administered GM-080 than in the placebo group after treatment (Table 6, *p* = 0.049). We also examined the effects of GM-080 administration on serum IgE production, skin sensitization, and serum IFN-γ levels at baseline and at the end of treatment (month 3). No changes in sera total IgE levels were observed either at baseline or at the end of treatment after 12 weeks among the groups (Table 7). We observed an increasing trend in IFN-γ levels of GM-080 consumption groups with middle and high doses (group B and group C, respectively) at the end of treatment compared to the placebo or low dose group (group A) (Table 7). However, the increases did not reach statistical differences. These data indicate the beneficial effects of GM-080 consumption in pediatric AR; however, these effects are not dosage-dependent but time-dependent. 

Additional well-designed clinical trials are warranted to identify the most effective dosage of GM-080.

## 4. Discussion

*Lactobacillus* spp. alleviate allergic diseases through different mechanisms. In mice, the oral administration of *Lactobacillus reuteri* for 9 days increased CD4^+^CD25^+^FoxP3^+^ regulatory T (Treg) cell numbers in the spleen; moreover, the adoptive transfer of these Treg cells from *L. reuteri*-treated mice reduced airway inflammation induced by antigen challenge [40]. Zhong et al. demonstrated that in OVA-sensitized rats, the administration of a mixture of probiotic genomic DNA derived from *L. rhamnosus* GG (LGG) and *Bifidobacterium longum* BB536 or that of a synthetic CpG-ODN reduced the production of Th2 cytokines and increased the Treg cell population in the spleen or mesenteric lymph nodes on the basis of increased toll-like receptor 9/nuclear factor kappa B activity [41]. CpG-ODN and LGG DNA containing TTTCGTTT, which is an IM, also demonstrated antiallergic potential in mice; specifically, they downregulated OVA-specific IgE production and increased systemic Th1 responses [42]. In the current study, the predicted IMs from BCRC 16100 did not display any Th1 cytokine induction capabilities and only IM5 in GM-080 exhibited IL-12 induction activity in mouse splenocytes (Table 2). Although the frequency of IMs in the genomes of probiotics may be associated with their antiallergic potential [43], the observed immunomodulatory activities of the predicted IMs warrant experimental examination. In the present study, we could not predict the relationship between the antiallergic activity of GM-080 and the associated changes in Th1 cytokine induction in vitro (Figure 2B,F) or the reduction in Th2 cytokine levels in vivo (Figure 5F). Given the Th1 cytokine induction activity of GM-080 IM5 and GM-080 ODN1 and ODN2 (Table 2), CpG-ODN and IMs in the GM-080 genome may underlie the beneficial effects of GM-080 against AHR.

Considering their IFN-γ induction activity in vitro (Figure 2C,D), LTA and PGN, two cell-wall components of GM-080, may be the key active ingredients involved in the improvement of AHR. Li et al. reported that PGN from *Lactobacillus acidophilus* inhibited IgE production and regulated Treg–Th17 balance, thus preventing β-lactoglobulin allergy [44]. Mat et al. indicated that treatment with LTA from *Staphylococcus aureus* reduced IL-5 production in peripheral blood mononuclear cells from patients with asthma [45], suggesting that LTA might have AHR-alleviating effects. However, the detailed molecular mechanisms underlying IFN-γ upregulation due to GM-080-derived LTA or PGN warrant further investigation.

In our WGS of GM-080 and BCRC 16100 (Table 1 and Appendix A), the undecaprenyl-phosphate galactose phosphotransferase gene (*rfbp*) and alpha-D-GlcNAc alpha-1,2-L-rhamnosyltransferase gene (*rgpAc*) involved in exopolysaccharide synthesis were noted only in GM-080. Wu et al. reported that exopolysaccharides from *B. longum* BCRC 14634 suppressed LPS-induced TNF-α production in J774A.1 macrophages [46]. Whether the immunomodulatory effect of GM-080-derived exopolysaccharides is responsible for the antiallergic activity of GM-080 remains unclear. In Treg cells, glutathione (GSH) loss can reduce suppressor function, thus inducing multiorgan autoimmunity. GSH-enriched yeast has been demonstrated to alleviate CCl_4_-induced liver damage in rats. In our analysis of the GM-080 genome, we noted the presence of *pepA*, an aminopeptidase that may be involved in GSH biosynthesis [47]. Considering that oxidative stress is a hallmark of asthma [48], GM-080 may be a GSH-enriched *Lactobacillus* that can improve the allergic condition.

In our in vitro analysis of the effects of GM-080 on Th1 cytokine (IL-12 and IFN-γ) production, the induction effect was unaffected by the MOI (Figure 2). This result indicates the presence of a therapeutic window of GM-080 dosage for the alleviation of airway allergic conditions. One animal study even confirmed that for GM-080, immunoregulatory function is determined by dehydrogenase activity [49]. In particular, the effectiveness of probiotics in the human environment may be affected by factors related to the internal microbial ecosystem, including intestinal colonization duration and changes in intestinal microbiota, after the consumption of probiotics [49]. The optimization of GM-080 dosage based on AHR severity warrants further investigation. In the OVA-induced asthma mouse model, we found that the oral consumption of GM-080 at the sensitization phase could reduce anti-OVA specific IgE at low doses (Figure 6A) or anti-OVA specific IgG2a at low and high doses (Figure 6C) but did not change anti-OVA specific IgG1 levels (Figure 6B). As a result of the general role of IgG1 or IgG2a for representing Th2 or Th1 responses, respectively [50], the data on unchanged anti-OVA specific IgG1 may not reflect the downregulation of Th2 responses by GM-080. Thus, we further examined OVA-induced IL-5 production after GM-080 consumption and found that it could be reduced at both doses of GM-080 (Figure 6F). Indeed, several studies related to the improvement in OVA-induced asthma in mouse models also reported the suppression of anti-OVA specific IgG2a [51,52]. The data of reducing serum OVA-specific-IgE and OVA-induced IL-5 production by mouse splenocytes in this study suggests that GM-080, at an optimal dose, could inhibit Th2 responses. 

In the current trial, the single live strain GM-080 was used, and its PAR-alleviating effect appeared to be significant only in terms of symptom (sneezing) relief (Table 4) and quality of life improvement (Table 6). Moreover, the effects of GM-080 on symptom (sneezing) relief were not dosage-dependent. In the present study, the elevated serum IFN-γ levels at the final visit were noted only in the patients who consumed a moderate amount of GM-080 at the dose of 2 × 10^9^ CFU per day (Table 7), indicating that IFN-γ induction may serve as a predictive factor for the selection of antiallergic probiotics. Lin et al. reported that in children aged 6–12 years, treatment with *L. paracasei* HF.A00232 at 5 × 10^9^ CFU/capsule combined with 5 mg of levocetirizine significantly ameliorated AR symptoms, including sneezing, itching nose, and swollen, puffy eyes [53]. Therefore, the effects of GM-080 in combination with antihistamines on AR symptoms should be investigated.

## 5. Conclusions

*L. paracasei* GM-080 induced Th1 cytokine production in mouse splenocytes and improved airway inflammation in an OVA-induced asthma mouse model with Th2 cytokine downregulation. This prospective, double-blind, placebo-controlled, randomized clinical trial on PAR showed the ameliorating effects of GM-080 on symptoms, such as sneezing, TNSS, and Investigator Global Assessment Scale score. Therefore, when given as a food supplement, GM-080 can alleviate AR in children.

## Figures and Tables

**Figure 1 cells-12-00768-f001:**
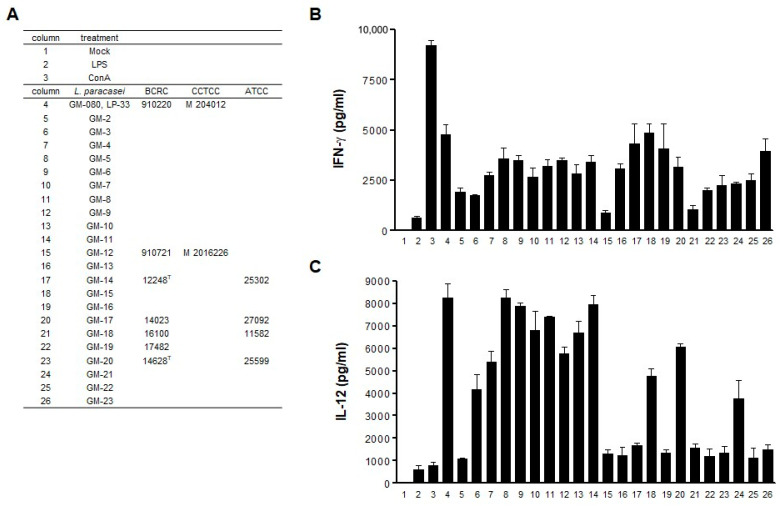
Th1 cytokine induction by GM-080 in mouse splenocytes. (**A**) List of *L. paracasei* strains used for coculture with mouse splenocytes. (**B**,**C**) Different strains of viable *L. paracasei* were used to coculture with mouse splenocytes at MOI = 10 for 48 h. The presence of IFN-γ (**B**) and IL-12 (**C**) in the culture supernatants was determined using commercial ELISA kits. LPS (1 µg/mL) or ConA (2 µg/mL) was used as the positive control.

**Figure 2 cells-12-00768-f002:**
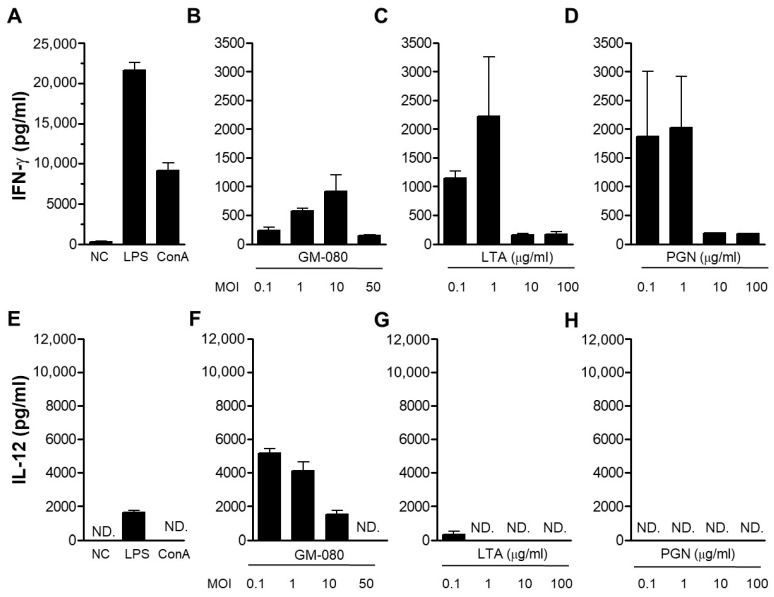
Effects of GM-080 preparations and cell wall compartments in Th1 cytokines production of mouse splenocytes. We seeded 4 × 10^5^ mouse splenocytes into a 48-well plate, followed by treatment with viable GM-080 preparation with the indicated MOI or cell wall compartments purified from GM-080 (LTA and PGN) for 48 h. The culture supernatants were then harvested, and INF-γ (**A**–**D**) or IL-12 (**E**–**H**) production was determined using commercial ELISA kits. LPS (1 µg/mL) or ConA (1 µg/mL) was used as the positive control. N.D., not detected.

**Figure 3 cells-12-00768-f003:**
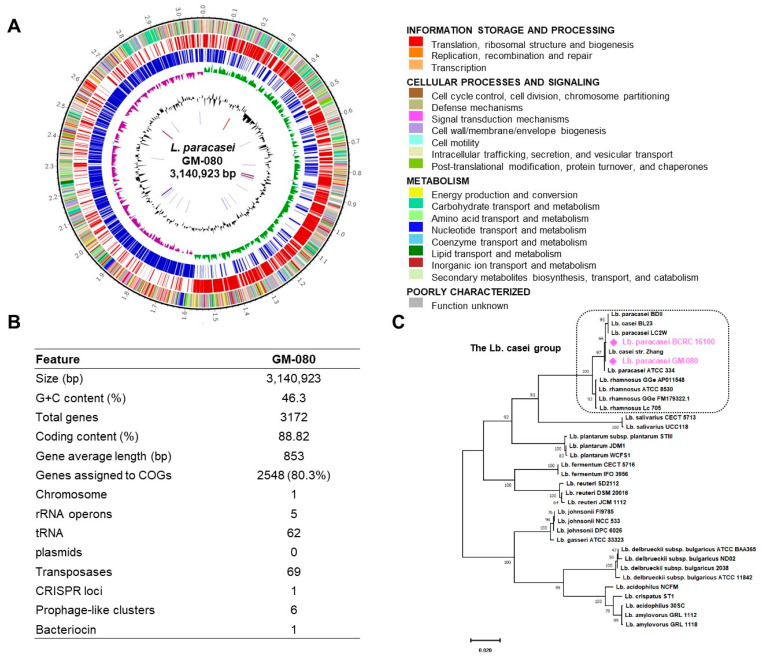
WGS analysis of GM-080. (**A**) Circular genome graph of GM-080 illustrated using Circos. (**B**) Summary of general features of GM-080 genome. (**C**) Maximum-likelihood tree of GM-080 and BCRC 16100 with 32 other known *Lactobacillus* strains.

**Figure 4 cells-12-00768-f004:**
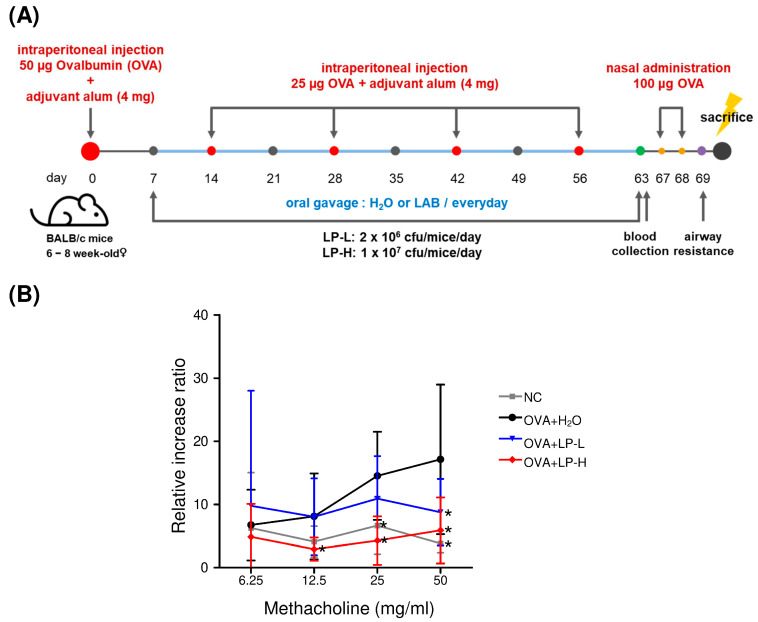
Effects of oral GM-080 on respiratory resistance in our OVA-induced AHR mouse model. (**A**) Illustration of the experimental procedure used for induction of OVA-induced AHR mouse model. Two dosages of GM-080 (LP-L, 2 × 10^6^ CFU/mouse/day; LP-H, 1 × 10^7^ CFU/mouse/day) were force-fed to mice through gavage. (**B**) Respiratory resistance was evaluated on the basis of Penh values under the inhalation of the indicated concentration of methacholine. NC, nontreated control; H_2_O, H_2_O gavage. * *p* < 0.05 compared with the H_2_O group.

**Figure 5 cells-12-00768-f005:**
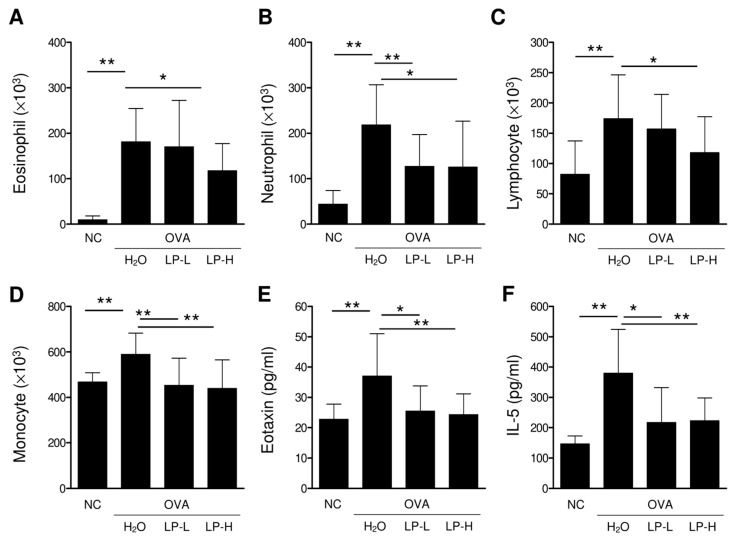
Effects of oral GM-080 on immune cell infiltration and eosinophilic cytokines in lungs of mice with AHR. BALF was collected 2 days after the final OVA challenge followed by centrifugation, and the cell pellet was collected. (**A**–**D**) Eosinophil (**A**), neutrophil (**B**), lymphocyte (**C**), and monocyte (**D**) infiltration was evaluated through microscopy after staining cells with Liu’s stain. (**E**,**F**) Eotaxin (**E**) or IL-5 (**F**) levels in BALF was determined using commercial ELISA kits. NC, nontreated control; H_2_O, H_2_O gavage; LP-L, GM-080 at 2 × 10^6^ CFU/mouse/day through gavage; LP-H, GM-080 at 1 × 10^7^ CFU/mouse/day through gavage. * *p* < 0.05, ** *p* < 0.01.

**Figure 6 cells-12-00768-f006:**
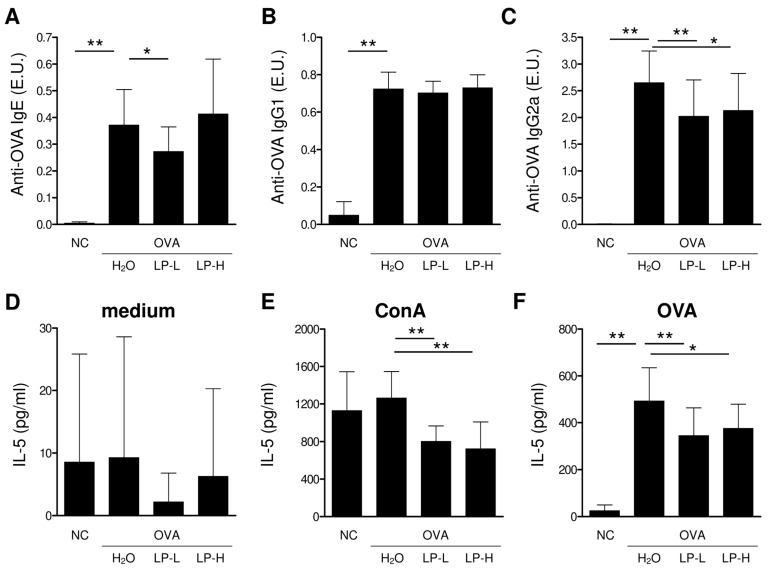
Effects of GM-080 consumption in OVA-specific antibodies or OVA-induced IL-5 production in an OVA-induced AHR mouse model. (**A**–**C**) Mice sera were collected 2 days after final theOVA challenge, and the presence of OVA-specific IgE (**A**), IgG1 (**B**), and IgG2a (**C**) was determined using ELISA. (**D**–**F**) Splenocytes were collected 2 days after the final OVA challenge and then seeded into a 48-well-plate at a density of 5 × 10^6^ cells/well to determine IL-5 production without treatment (**D**) or after treatment with 2 µg/mL ConA (**E**) or 100 µg/mL OVA (**F**). NC, nontreated control; H_2_O, H_2_O gavage; LP-L, GM-080 at 2 × 10^6^ CFU/mouse/day through gavage; LP-H, GM-080 at 1 × 10^7^ CFU/mouse/day through gavage. * *p* < 0.05, ** *p* < 0.01.

**Figure 7 cells-12-00768-f007:**
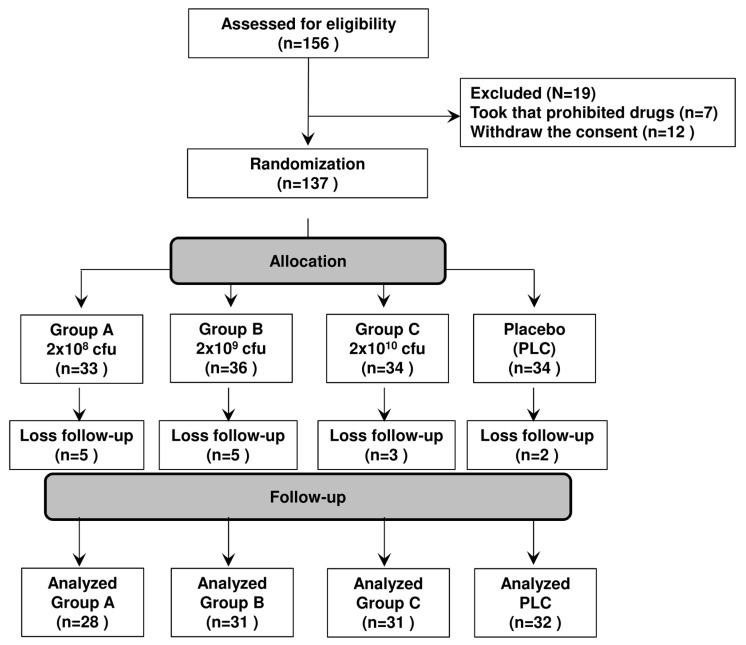
Enrollment of children with AR in our double-blind, randomized, placebo-controlled trial. The inclusion and exclusion criteria are described in the Section 2.

**Table 1 cells-12-00768-t001:** COG functional categories of GM-080 and BCRC 16100.

COG Functional Categories ^a^	Same Gene No.	Unique Genes in GM-080 (Total No.)	Unique Genes in BCRC 16100 (Total No.)
Cell wall/membrane/envelope biogenesis (M)	72	*rfbP*, *rgpAc*, *mprF (3)*	*kdsD*, *tuaG*, *tagF*, *ywqC*, *tagE*, *gtf1 (6)*
Replication, recombination, and repair (L)	89	*recT*, *pi112*, *tnpA1*, *tnp1216*, *cas2*, *cas1*, *cas9*, *int3*, *pi346*, *tnpR*, *is18*, *yqaJ*, *rusA (13)*	*-*
Posttranslational modification, protein turnover, and chaperones (O)	44	*gst (1)*	*-*
Carbohydrate transport and metabolism (G)	139	*agaD*, *kduI*, *kdgK*, *ahaA*, *xylP*, *lacE*, *lacG*, *lacF (8)*	*pts32BC*, *gatY*, *mnaA (3)*
Amino acid transport and metabolism (E)	132	*dppA*, *pepA*, *yxeO (3)*	*-*
Coenzyme transport and metabolism (H)	53	-	*pdxA (1)*
Inorganic ion transport and metabolism (P)	92	*feoA*, *ytmL (2)*	*kdgT*, *sfuB*, *fbpC (3)*
Secondary metabolites biosynthesis, transport, and catabolism (Q)	15	*kduD (1)*	*-*
Intracellular trafficking, secretion, and vesicular transport (U)	36	*chaT1*, *clpP (2)*	*secY2*, *secA2 (2)*

^a^ COG, Clusters of Orthologous Gene. The analysis of COG categories was done by National Center for Biotechnology Information (https://www.ncbi.nlm.nih.gov/research/cog/, access on 1 November 2021).

**Table 2 cells-12-00768-t002:** Immunoregulatory effects of putative IM and CpG-containing ODN ^a^ in GM-080 and BCRC 16100.

Code	Core Sequence	Strain	Sequence	IFN-r (pg/mL) ^b^	IL-12 (pg/mL) ^c^
IM3	TCAAGCTTGA		TCAAGCTTGA	ND ^d^	ND
IM4	TCAAGCTTGA	GM-080	CAAGCGTCAAGCTTGAATGA	ND	ND
IM5		GM-080	AAAAATTCAAGCTTGATAGT	ND	609.8 ± 227.69
IM6		GM-080	CCATCGTCAAGCTTGACTTG	ND	ND
IM7		GM-080	CCCTAATCAAGCTTGATTAA	ND	ND
IM8		BCRC 16100	GCAGCTTCAAGCTTGAAAAA	ND	ND
IM9		BCRC 16100	CCGGCCTCAAGCTTGAATTG	ND	ND
IM10		BCRC 16100	TTTCATTCAAGCTTGACGCT	ND	ND
IM11		BCRC 16100	CCTTAATCAAGCTTGATTAG	ND	ND
ODN1	GACGATCGTC	GM-080	GCTTGACGATCGTCTCTGGA	ND	38.8 ± 65.2
ODN2	ACGACGTCGT	GM-080	GGTCACGACGTCGTTTACAAA	490 ± 272.74	46.1 ± 32.88
ODN3	GACGATCGTC	BCRC 16100	AATTGACGATCGTCTAATTC	ND	ND
ODN4		BCRC 16100	TGTCGACGATCGTCGTCTGT	ND	ND
ODN5		BCRC 16100	CAGAGACGATCGTCAAGCGA	77.381 ± 5.4	ND
ODN6	ACGACGTCGT	BCRC 16100	CGTCACGACGTCGTGACCGGC	ND	ND

^a^ ODN, oligonucleotides. ^b^ Treatment concentration was 0.125 µM. Data are presented as means ± standard deviations. ^c^ Treatment concentration was 1 µM. Data are presented as means ± standard deviations. ^d^ ND, not detected.

**Table 3 cells-12-00768-t003:** Demographic and baseline characteristics of patients ^a^.

Characteristics	A Group (*n* = 28)	B Group (*n* = 31)	C Group (*n* = 31)	PLC Group (*n* = 32)	*p*
Male ^b^	20 (71.4%)	18 (58.1%)	18 (58.1%)	21 (65.6%)	0.660
Age (year) ^c^	8.25 (2.78)	8.74 (2.77)	8.42 (2.55)	8.56 (2.95)	0.917
Sneezing ^c^	1.69 (0.77)	1.61 (0.67)	1.49 (0.79)	1.75 (0.64)	0.527
Rhinorrhea ^c^	1.83 (0.55)	1.81 (0.68)	1.69 (0.70)	1.95 (0.60)	0.435
Nasal pruritus ^c^	1.73 (0.70)	1.72 (0.53)	1.69 (0.67)	1.70 (0.79)	0.997
Nasal congestion ^c^	2.10 (0.63)	1.97 (0.66)	2.04 (0.71)	2.04 (0.61)	0.880
TNSS ^c^	7.34 (1.44)	7.11 (1.74)	6.92 (1.57)	7.44 (1.56)	0.559
Total serum IgE (kU/L) ^c^	652.58 (1366.19)	547.66 (530.99)	614.81 (661.08)	405.68 (384.19)	0.642
Combine with asthma ^b^	11 (39.3%)	14 (45.2%)	11 (35.3%)	12 (37.5%)	0.878
Combine with atopic dermatitis ^b^	12 (42.9%)	11 (35.5%)	8 (25.8%)	11 (34.4%)	0.590
Combine with conjunctivitis ^b^	5 (17.9%)	6 (19.4%)	5 (16.1%)	9 (28.1%)	0.648
Allergic sensitization ^b^					
Mite (Df)	23 (82.1%)	25 (80.6%)	20 (64.5%)	25 (78.1%)	0.350
Mite (Dp)	26 (92.9%)	30 (96.8%)	28 (90.3%)	31 (96.9%)	0.624
Cockroach	6 (21.4%)	7 (22.6%)	10 (32.3%)	5 (15.6%)	0.470
Animal dander (Cats)	6 (21.4%)	4 (12.9%)	7 (22.6%)	7 (21.9%)	0.749
Animal dander (Dogs)	7 (25.0%)	6 (19.4%)	7 (22.6%)	9 (28.1%)	0.869
Mold	1 (3.6%)	2 (6.5%)	2 (6.5%)	1 (3.1%)	0.887

^a^ Severity of each symptom was measured on a 4-point scale: 0 = absent; 1 = mild; 2 = moderate; 3 = severe. Scores for each symptom were added to obtain the TNSS. ^b^ Data are presented as *n* (%). ^c^ Data are presented as mean (standard deviation).

**Table 4 cells-12-00768-t004:** AR symptom scores at baseline and follow-up visits among enrolled patients ^#^.

Subscale	Examination	A Group (*n* = 28)	B Group (*n* = 31)	C Group (*n* = 31)	PLC Group (*n* = 32)	*p*
Sneezing	Visit 2	1.69 (0.77)	1.61 (0.67)	1.49 (0.79)	1.75 (0.64)	0.527
	Visit 3	1.61 (0.71)	1.20 (0.57) ^a^	1.43 (0.83)	1.67 (0.67)	0.045 ^e^
	Visit 4	1.68 (0.75)	1.19 (0.70) ^a^	1.41 (0.83)	1.40 (0.65)	0.090
	Visit 5	1.51 (0.74)	1.02 (0.66) ^a^	1.35 (0.96)	1.50 (0.66)	0.049 ^e^
	Visit 6	1.33 (0.76)	1.08 (0.68) ^a^	1.06 (0.82) ^ab^	1.56 (0.82)	0.033 ^e^
Rhinorrhea	Visit 2	1.83 (0.55)	1.81 (0.68)	1.69 (0.70)	1.95 (0.60)	0.435
	Visit 3	1.71 (0.67)	1.73 (0.83)	1.51 (0.86)	1.95 (0.75)	0.170
	Visit 4	1.69 (0.71)	1.42 (0.80) ^a^	1.39 (0.76)	1.65 (0.66)	0.252
	Visit 5	1.37 (0.73)	1.20 (0.72) ^ab^	1.24 (1.00)	1.59 (0.69)	0.203
	Visit 6	1.42 (0.81)	1.11 (0.67) ^ab^	1.24 (0.99)	1.44 (0.75)	0.320
Nasal pruritus	Visit 2	1.73 (0.70)	1.72 (0.53)	1.69 (0.67)	1.70 (0.79)	0.997
	Visit 3	1.54 (0.83)	1.42 (0.73)	1.35 (0.67)	1.65 (0.80)	0.415
	Visit 4	1.43 (0.81)	1.34 (0.78) ^a^	1.24 (0.59) ^a^	1.51 (0.72)	0.466
	Visit 5	1.24 (0.64) ^a^	1.09 (0.64) ^a^	1.11 (0.78) ^a^	1.47 (0.75)	0.123
	Visit 6	1.08 (0.77) ^a^	1.08 (0.80) ^a^	1.09 (0.75) ^a^	1.33 (0.89)	0.520
Nasal congestion	Visit 2	2.10 (0.63)	1.97 (0.66)	2.04 (0.71)	2.04 (0.61)	0.880
	Visit 3	1.87 (0.74)	1.87 (0.82)	1.86 (0.88)	2.07 (0.76)	0.663
	Visit 4	1.87 (0.81)	1.60 (0.78)	1.72 (0.64)	1.84 (0.72)	0.476
	Visit 5	1.33 (0.71) ^ab^	1.35 (0.88) ^a^	1.47 (0.89) ^a^	1.77 (0.84)	0.154
	Visit 6	1.29 (0.73) ^abc^	1.15 (0.88) ^abc^	1.19 (0.90) ^abc^	1.48 (0.84) ^a^	0.414
TNSS	Visit 2	7.34 (1.44)	7.11 (1.74)	6.92 (1.57)	7.44 (1.56)	0.559
	Visit 3	6.74 (2.01)	6.23 (2.21)	6.15 (2.51)	7.35 (2.24)	0.135
	Visit 4	6.68 (2.45)	5.54 (2.50) ^a^	5.77 (1.95) ^a^	6.40 (2.05)	0.174
	Visit 5	5.44 (2.13) ^a^	4.67 (2.41) ^ab^	5.18 (2.97) ^a^	6.33 (2.36)	0.066
	Visit 6	5.12 (2.44) ^abc^	4.42 (2.39) ^ab^	4.57 (3.00) ^abc^	5.56 (2.49)	0.286

^#^ Severity of each symptom was measured on a 4-point scale: 0 = absent; 1 = mild; 2 = moderate; 3 = severe. Scores for each symptom were added to obtain the TNSS. Data were presented as mean (standard deviation). ^a^
*p* < 0.05, intragroup comparisons (visit 6 vs. visit 2, visit 5 vs. visit 2, visit 4 vs. visit 2, visit 3 vs. visit 2 in each group). ^b^
*p* < 0.05, intragroup comparisons (visit 6 vs. visit 3, visit 5 vs. visit 3, visit 4 vs. visit 3). ^c^
*p* < 0.05, intragroup comparisons (visit 6 vs. visit 4, visit 5 vs. visit 4). ^e^
*p* < 0.05, intragroup comparisons (difference between the four groups).

**Table 5 cells-12-00768-t005:** Differences in sneezing, rhinorrhea, nasal pruritus, nasal congestion, and TNSS among the four groups after five visits based on the GEE model.

	A Group (*n* = 28)		B Group (*n* = 31)		C Group (*n* = 31)		PLC Group (*n* = 32)		*p*
Subscale	*n*	Value (95% CI ^a^)	*n*	Value (95% CI)	*n*	Value (95% CI)	*n*	Value (95% CI)	
Sneezing	28	0.028 (−0.321, 0.377)	31	−0.229 (−0.536, 0.077)	31	−0.250 (−0.604, 0.104)	32	Referent	0.041 ^b^
Rhinorrhea	28	−0.014 (−0.404, 0.376)	31	−0.332 (−0.677, 0.012)	31	−0.202 (−0.627, 0.223)	32	Referent	0.104
Nasal pruritus	28	−0.258 (−0.671, 0.155)	31	−0.250 (−0.662, 0.161)	31	−0.242 (−0.643, 0.159)	32	Referent	0.414
Nasal congestion	28	−0.192 (−0.583, 0.199)	31	−0.330 (−0.750, 0.089)	31	−0.289 (−0.713, 0.136)	32	Referent	0.441
TNSS	28	−0.436 (−1.663, 0.791)	31	−1.142 (−2.328, 0.044)	31	−0.983 (−2.324, 0.357)	32	Referent	0.070

^a^ CI, confidence interval. ^b^
*p* < 0.05, intergroup comparisons (difference between four groups).

**Table 6 cells-12-00768-t006:** Investigator Global Assessment Scale scores at baseline and follow-up visits among the four groups ^a^.

Subscale	Examination	A Group (*n* = 28)	B Group (*n* = 31)	C Group (*n* = 31)	PLC Group (*n* = 32)	*p*-Value 4 Groups
Global Assessment	Visit 3	2.43 (0.88)	2.42 (0.62)	2.58 (0.56)	2.47 (0.72)	0.795
	Visit 4	2.61 (0.63)	2.61 (0.67)	2.71 (0.69)	2.44 (0.76)	0.471
	Visit 5	3.00 (0.61) ^bc^	2.94 (0.81) ^b^	2.77 (0.76)	2.53 (0.84)	0.083
	Visit 6	2.96 (0.74) ^c^	2.97 (0.71) ^b^	3.16 (0.86) ^d^	2.59 (0.91)	0.049 ^e^

^a^ Data are presented as mean (standard deviation). ^b^
*p* < 0.05, intragroup comparisons (visit 6 vs. visit 3, visit 5 vs. visit 3, visit 4 vs. visit 3 in each group). ^c^
*p* < 0.05, intragroup comparisons (visit 6 vs. visit 4, visit 5 vs. visit 4). ^d^
*p* < 0.05, intragroup comparisons (visit 6 vs. visit 5). ^e^
*p* < 0.05, intragroup comparisons (difference between the four groups).

**Table 7 cells-12-00768-t007:** Serum biomarker and sensitization levels at baseline and at the end of treatment ^#^.

Biomarker	Examination	A group (*n* = 28)	B group (*n* = 31)	C group (*n* = 31)	PLC group (*n* = 32)	*p*
Total IgE (kU/L) ^a^	Baseline	652.58 (1366.19)	547.66 (530.99)	614.81 (661.08)	405.68 (384.19)	0.642
	Visit 6	648.73 (1466.52)	449.69 (469.01)	599.39 (677.36)	419.52 (529.22)	0.833
IFN-γ (ng/mL) ^a^	Baseline	436.96 (366.48)	494.73 (760.74)	302.30 (442.22)	383.25 (498.97)	0.331
	Visit 6	467.10 (915.50)	1042.03 (3006.39) ^b^	740.28 (2020.84)	515.89 (1830.42)	0.480

^#^ Measurements were performed in month 3. ^a^
*p* < 0.05, intragroup comparisons (visit 6 vs. baseline in each group). ^b^ Data are presented as means (standard deviations).

## Data Availability

The whole genome sequencing data of BCRC 16100 were available in GenBank^®^ (National Center for Biotechnology Information, National Library of Medicine, National Institutes of Health, USA) with the accession number of GCA_022588775.1. The nucleotide sequences of GM-080 have been submitted into BioProject in NCBI with an accession number of PRJNA824946.

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
