# Peer review of "Lacticaseibacillus paracasei GM-080 Ameliorates Allergic Airway Inflammation in Children with Allergic Rhinitis: From an Animal Model to a Double-Blind, Randomized, Placebo-Controlled Trial"

_cells, 2023, doi:10.3390/cells12050768_

Round 1

Reviewer 1 Report

The authors evaluated the efficacy and safety of Lacticaseibacillus paracasei GM-080 (which was originally detected from 23 strains by in vitro study) by murine model of asthma and a clinical study with pediatric allergic rhinitis patients. The manuscript is generally well-written. However, I have concerns as follows.

Major comments>

L66-67 There are many publications on the effectiveness of probiotics in asthma. Please search “probiotic* allergy airway inflammation” at PubMed.

L217. The target disease was allergic rhinitis. What is the reason to evaluate food allergen sensitization in this patient group?

L321 According to the protocol, GM-080 was administered at sensitization phase. What if it were administered during the challenge phase which is more therapeutic approach?

L333-336 How do the authors interpret OVA-specific antibodies? OVA-specific IgE or IdG1 correlate with Th2 responses and OVA-specific IgG2a correlates with Th1.

Please also describe Th1 and regulatory T cell-related cytokine productions (e.g. IL-12 and IL-10) from splenocytes of the murine model. These results would be important to interpret the antibody data.

L207-208, L379, L496, L511 TNSS is for measuring the nasal symptoms, not the quality of life. For the evaluation of the quality of life, rhinoconjunctivitis Quality of Life Questionnaire (RQLQ) could be used.

L384 Table 6. Investigator global assessment was not described in the methods session. Please describe the method in detail in the methods session.

L389-390 L500 Table 7. Please check the p-value for IFNgamma at visit 6 which is not significant (p=0.480)

Minor comment >

L42: may use -> maybe used

L219-221. Please list the names of allergens that were evaluated by UniCAP.

Reviewer 2 Report

Good study; good study design and rationale 

Line 179-181: Inclusion criteria 

How to diagnosed PAR

MAST score=2   ?? please define meaning

how to diagnose rhinitis medica mentosa, why included patient with syndrome

Reviewer 3 Report

This is an interesting article, conducted either in patients and mice, written to demonstrate the usefulness of probiotics in the management of allergic diseases.

The authors selected Lacticaseibacillus paracasei strain GM-080, known to alleviate various allergic diseases, as probiotic target and examined its therapeutic effects in both asthmatic mouse and a double-blind, prospective, randomized, placebo-controlled study in children with AR.

The various methods used are appropriate to the purpose of the work and are accurately described.

The article is generally well written and structured. The introduction provides sufficient background and include all relevant references. However, for greater clarity I suggest dividing the abstract into sections and adding a short “take on message” section.

Below few observations:

1.     improve the resolution of Figure 3

2.     fix table 1 which is shifted

3.     fix table 2 which is shifted too

4.     figure 4a: improve resolution and magnification

Also, the quality of the writing could have been much better.

However, I believe the article demonstrates high scientific value and is worth reading.

Author Response

Response to Reviewer 3 Comments

Point 1: The article is generally well written and structured. The introduction provides sufficient background and include all relevant references. However, for greater clarity I suggest dividing the abstract into sections and adding a short “take on message” section.

Response 1: We thank the comments from the reviewer. The abstract section now has been revised by dividing into sections including Background, Methods, Results, and Conclusion. The conclusion section could be set as “take on message” that suggested by the reviewer.

Point 2: Below few observations:

  1. improve the resolution of Figure 3; 2. fix table 1 which is shifted
  2. fix table 2 which is shifted too; 4. figure 4a: improve resolution and magnification

Response 2: These points have been corrected in the revised manuscript.

Point 3: Also, the quality of the writing could have been much better.

Response 3: The revised manuscript has been edited by a native speaker and we include the certificate as a supplementary material for reviewing.

In general, we appreciate and thank the positive feedback and comments from the reviewer.
